# Evaluation of a Navigated 3D Ultrasound Integration for Brain Tumor Surgery: First Results of an Ongoing Prospective Study

**Danilo Aleo** [1], **Ziad Elshaer** [2], **Andreas Pfnür** [2], **Patrick J. Schuler** [3,4], **Marco Maria Fontanella** [1], **Christian Rainer Wirtz** [2,4], **Andrej Pala** [2,*] and **Jan Coburger** [2,4]

1   Neurosugery, Department of Medical and Surgical Specialties, Radiological Science and Public Health, University of Brescia, 25121 Brescia, Italy
2   Department of Neurosurgery, Ulm University, Lindenallee 2, 89312 Günzburg, Germany
3   Department of Otorhinolaryngology, Head and Neck Surgery, Ulm University Medical Center, 89081 Ulm, Germany
4   Surgical Oncology Ulm, i2SOUL Consortium, Ulm University Medical Center, 89081 Ulm, Germany
*   Correspondence: andrej.pala@uni-ulm.de; Tel.: +49-8221-9628-866

**Abstract:** The aim of the study was to assess the quality, accuracy and benefit of navigated 2D and 3D ultrasound for intra-axial tumor surgery in a prospective study. Patients intended for gross total resection were consecutively enrolled. Intraoperatively, a 2D and 3D iUS-based resection was performed. During surgery, the image quality, clinical benefit and navigation accuracy were recorded based on a standardized protocol using Likert's scales. A total of 16 consecutive patients were included. Mean ratings of image quality in 2D iUS were significantly higher than in 3D iUS ($p < 0.001$). There was no relevant decrease in rating during the surgery in 2D and 3D iUS ($p > 0.46$). The benefit was rated 2.2 in 2D iUS and 2.6 in 3D iUS ($p = 0.08$). The benefit remained stable in 2D, while there was a slight decrease in the benefit in 3D after complete tumor resection ($p = 0.09$). The accuracy was similar in both (mean 2.2 $p = 0.88$). Seven patients had a small tumor remnant in intraoperative MRT (mean 0.98 cm$^3$) that was not appreciated with iUS. Crucially, 3D iUS allows for an accurate intraoperative update of imaging with slightly lower image quality than 2D iUS. Our preliminary data suggest that the benefit and accuracy of 2D and 3D iUS navigation do not undergo significant variations during tumor resection.

**Keywords:** intraoperative 3D ultrasound; navigation; glioma surgery; extent of resection; image quality





## 1. Introduction

The most recent scientific evidence demonstrates how maximal safe resection positively influences the prognosis of patients with high-grade gliomas. The extent of resection (EOR) is an independent prognostic factor associated with an improved clinical outcome [1,2]. More generally, EOR is a significant predictor in terms of progression free survival and overall survival for both high-grade (WHO III–IV) and low-grade (WHO II) glial tumors [3–5]. Similarly, gross total resection is reflected in longer overall survival in patients with single brain metastases [6]. For these reasons, image-guided surgery using neuro-navigation, introduced into the routine of neurosurgical practice in the 1990s, has become the standard for brain tumor surgery. However, the initial enthusiasm for such surgery was dampened by the immediate evidence of changes that occurred in the intra-operative anatomical space during the surgical procedure, including the phenomenon of "brain shift", which inevitably reflected the lack of accuracy of preoperatively acquired images [7–9]. Therefore, the need for intraoperative image updating has become progressively essential in order for surgeons to be able to cope with the paraphysiological dynamics of intracranial anatomical change as a result of a combination of CSF loss, an increase in intracranial pressure and consequent cerebral swelling, gravity, the position of the head and the use of hyperosmotic drugs. Among the current intraoperative imaging

technologies, the current gold standard is intraoperative MRI, which provides the best diagnostic definition at the expense of a high cost, an increased operating time and the interruption of the surgical workflow. Intraoperative ultrasound (iUS) is a valid and useful alternative, providing real-time information regarding the extent of resection, which can be used to update the neuro-navigation system [10]. iUS, used in neurosurgery since the 1980s [11,12], has a number of limitations, among them a reduced graphic definition when compared with CT and MRI scans, not to mention the variability of the interpretation of the operator-dependent image.

Moreover, the lack of familiarity among neurosurgeons with this technology due to the loss of three standard orthogonal planes and generally to a lack of specific training must be recognized [13,14]. Furthermore, the increase of artifacts as the surgical dissection progresses makes the interpretation of the results gradually more complex [15]. The advantages of iUS include its ease of set-up, its low operating costs, the possibility of providing images in real time, the absence of ionizing radiation, the possibility of using it several times in the same operation and the ability to depict almost all intra- and extra-axial lesions [16]. In recent decades, technological advancements in terms of graphic processing have significantly improved the quality of ultrasound images. The possibility of combining iUS with neuro-navigation technology has provided a synergistic benefit, allowing for reliable image guidance in real time [17,18]. Our group has shown that the use of linear array intraoperative ultrasound (15–17 MHz, lioUS) is associated with improved image quality, accuracy and sensitivity when detecting residual tumors when compared with cioUS (conventional intraoperative ultrasound) [19]. The system, based on higher resolution linear probes, showed an accuracy comparable to iMRI for both high and low grade tumors [19,20]. However, the wide-angle field of view and the higher penetration depth of cioUS means that it still has benefits with regards to orientation, tumor detection and the control of brain shift, especially during the pre-resection stages [19]. The benefits of the two types of probes, in agreement with Moiyadi [21], therefore appear to be complementary to each other. In fact, while cioUS provides better insonation of deeper lesions thanks to the greater wavelength and the curvature of the footprint, which allows for a wider view, lioUS has a better resolution for superficial lesions and a limited field of vision in depth [21]. Furthermore, the possibility of 3D volumetric reconstruction, obtained from the sum of multiple 2D scans generated by the movement of the conventional probe on the field of interest, can help to overcome the problem of spatial orientation and anatomical changes caused by brain shift [22,23]. The 3D reconstruction of iUS combined with neuro-navigation was pioneered by Unsgard et al. when developing the Sonowand system [24]. Such advances have been shown to be useful in detecting residual tumor [25]. Technology has improved over the years both in terms of ultrasound hardware as well as software reconstruction. While the Sonowand system is no longer available on the market, other systems for 3D reconstruction have arisen, such as ultrasound navigation software by Brainlab®, which we evaluated in the current study. The objective of this prospective study was to assess the image quality, surgical benefit in terms of tumor orientation, control and navigational accuracy of navigated 2D and 3D iUS in intra-axial tumor surgery.

## 2. Materials and Methods

A prospective study protocol was conducted in which adult patients with preoperative radiological suspicion of a potentially malignant intra-axial brain tumor, in particular brain metastasis, low- and high-grade brain glioma (grade II, III or IV, according to the WHO 2021 classification), were prospectively enrolled.. Inclusion criteria were a potential resection of the lesion of at least 95% (gross total resection—GTR) and patient's informed consent. Patients who underwent first surgery or surgery for tumor recurrence were included (Figure 1).

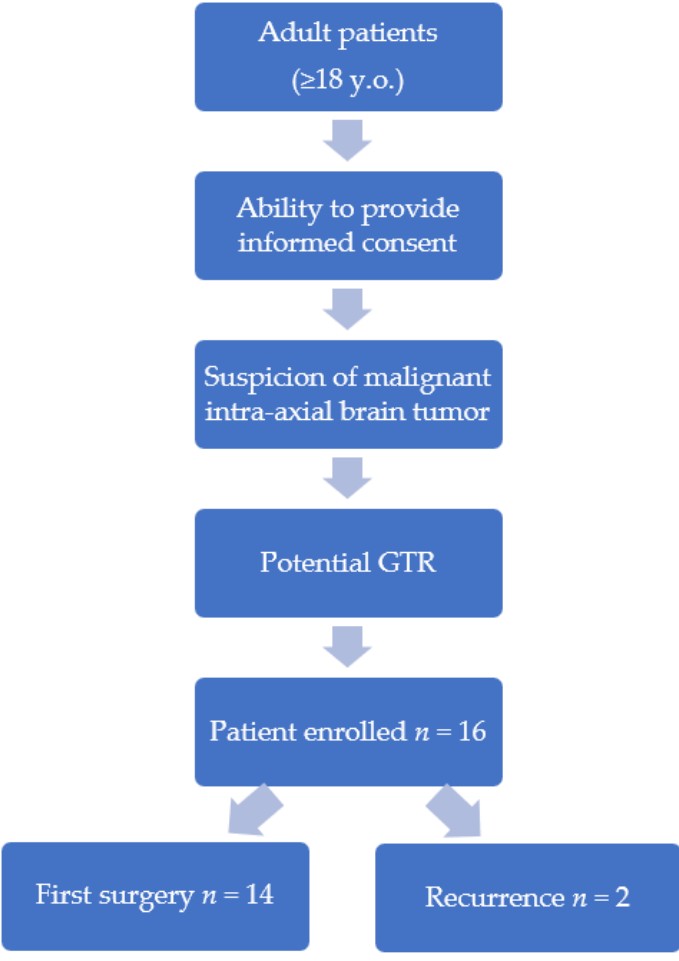

**Figure 1.** Flow chart designed for patient enrollment.

A surgical resection guided by intraoperative 2D and 3D ultrasound (BK 5000) integrated with the neuro-navigation system (Brainlab AG, Munich, Germany) was performed for each case. Data was collected on the image quality, surgical benefit and navigational accuracy of both types of techniques according to a questionnaire provided intraoperatively to the operating surgeon. Ratings were collected at fixed times during the operation, in particular at the opening of the dura mater, at the time of 50% presumed tumor mass resection and finally at the presumed achievement of complete white light resection. For this purpose, the surgeon's assessment was requested by using a Likert scale (1 = very good, 2 = good, 3 = fair, 4 = sufficient, 5 = bad, 6 = very bad). Furthermore, a 5–13 MHz micro-convex transducer was used during all surgical steps. The 2D scan did not follow a specific orientation but varied according to tumor location. The 3D dataset was acquired by a single wide sweep following the orientation that ensured greater tumor coverage through craniotomy. After the alleged GTR, intraoperative high-field magnetic resonance imaging (iMRI; 1.5 T Magnetom Espree, Siemens AG Healthcare Sector, Erlangen, Germany) was performed to assess the achieved extent of the resection. A follow-up postoperative MRI was performed in all patients within 48 h after the surgery. The preoperative volumes of the lesions and intra- and postoperative volumes of any residual tumors were then measured in $cm^3$ by means of MR images using the Elements® software (Brainlab, Munich, Germany). The operating times and interruption times of the intervention during the iMRI, the patient's personal data as well as the histological and molecular indicators of the diagnosis were also collected. The pre- and postoperative clinical data, both at the time of admission and at least three days after surgery, were obtained through the use of the Neurological Assessment in Neuro-Oncology (NANO) and the Karnofsky Performance Status (KPS). The statistical analysis of the descriptive qualitative evaluations, was carried out through non-parametric

tests. We used Wilcoxon test to compare the results of 2D and 3D iUS in the various fields of investigation and through the Sign test, to compare the coresponding ratings at different times throughout the surgery. A *p*-value of <0.05 was considered statistically significant. All statistical procedures were performed using IBM® SPSS® Statistics software, version 28.0 (IBM Corp, Armonk, NY, USA).

## 3. Results

From October 2021 to March 2022, 16 patients who met the inclusion criteria were consecutively enrolled in the study. The average age was 56 years. Six patients had glioblastoma WHO IV, and one of these was a recurrent glioblastoma. One patient was diagnosed with WHO IV astrocytoma IDH-mutant. Two patients were diagnosed with WHO III IDH-mutant astrocytoma, with one of these being a recurrence, and two others with WHO II IDH-mutant astrocytoma. Two patients had an oligodendroglioma (WHO II and III). Three patients with single brain metastases were included. The primary tumors were gastric adenocarcinoma, renal cell carcinoma and cutaneous melanoma (Table A1).

The calculated mean preoperative tumor volume was 32 cm$^3$. The hemisphere where the most frequent lesions were located was the right (11 cases, compared with 5 cases in the left). Nine cases involved the frontal lobe, five the temporal, one the parietal and one the occipital lobe. In 10 cases, the tumor occurred in a non-eloquent area, while in 6 cases the areas of speech (3), sight (2) and motor (1) were involved. The average operating time was 308 min, while the time needed to perform the iMRI until the resumption of surgery was on average 64 min (Table A1). After iMRI, GTR was achieved in all patients, although seven out of sixteen patients had a small remnant on iMRI (mean 0.98 cm$^3$) that was not appreciated with iUS. Figure 2 shows how the median of the overall values in terms of the quality, benefit and accuracy of both 2D iUS and 3D iUS stands at "2" according to the Likert scale used, corresponding with a "good" rating. The average ratings in terms of 2D image quality are significantly higher than those for 3D quality (1.60 vs. 2.31, *p* < 0.001). Otherwise, there are no statistically significant differences between the averages in terms of benefit and accuracy regarding the two ultrasound modalities (respectively 2.20 vs. 2.60, *p* = 0.087 for benefit; 2.21 vs. 2.25, *p* = 0.882 for accuracy).

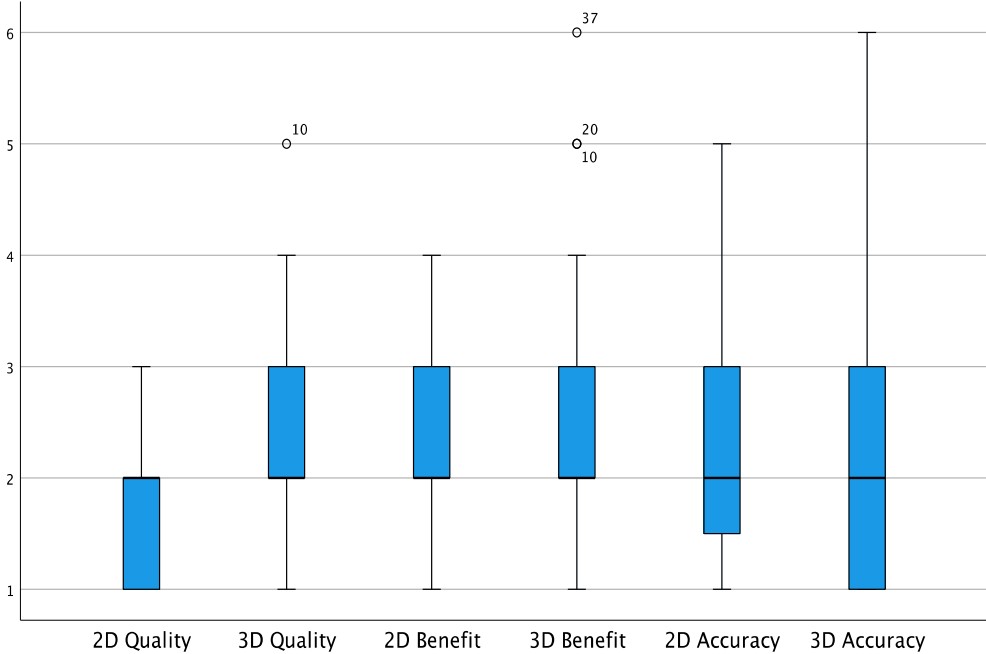

**Figure 2.** Box plot showing the overall values for quality, benefit and accuracy (Likert scale). White circles = low potential outlier (more than 1.5 inter-quartile range at most 3 inter-quartile range.)

When analyzing the relationship between the ratings assigned to the quality of intraoperative ultrasound images in 2D and 3D in the different stages of resection in detail, a significant difference was confirmed in favor of 2D images in the scans performed after opening the dura ($p = 0.046$), at the halfway point of surgical resection ($p = 0.016$) and at the end of resection ($p = 0.010$) (Figure 3). The Sign test, used to compare any variation in image quality between the various phases of the surgery, did not show any significant decrease, both for 2D and for 3D.

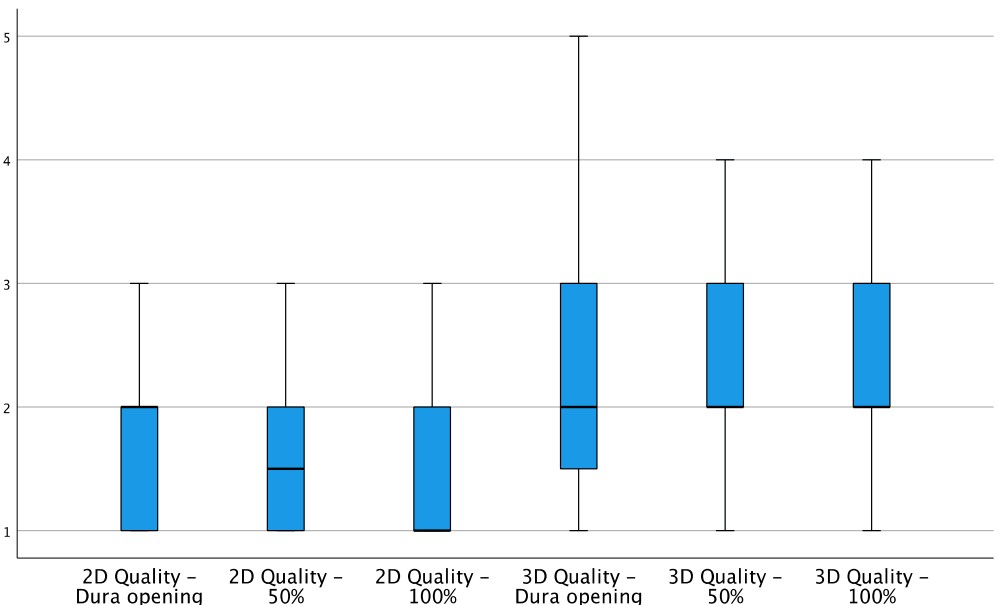

**Figure 3.** Box plot showing 2D and 3D quality variations during the different stages of resection.

The subjective benefit for the surgeon when using iUS does not appear to vary significantly during the course of the resection with regard to 2D; however, there was a slight worsening of 3D ratings at the completion of the resection (Figure 4), although this variation did not reach a statistically significant difference ($p = 0.092$).

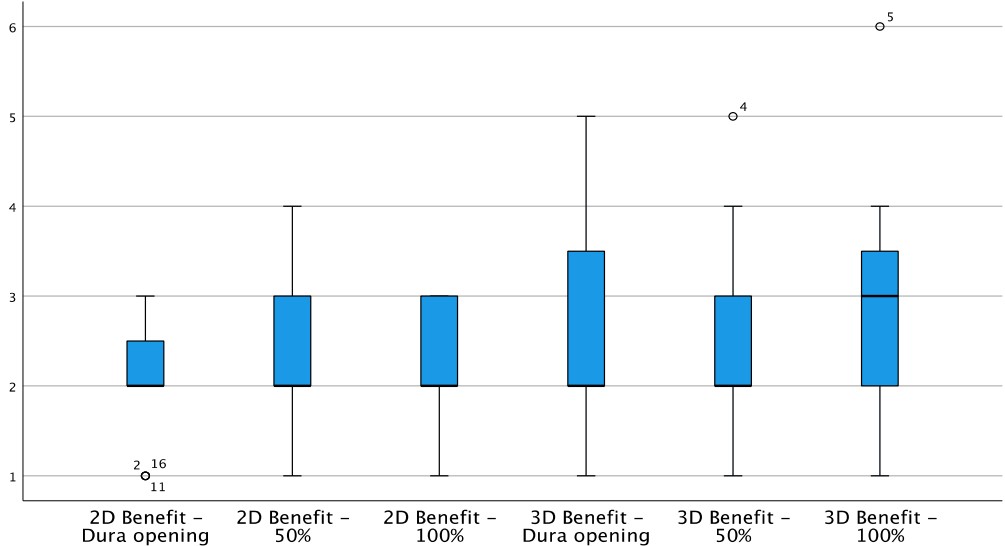

**Figure 4.** Box plot showing 2D and 3D benefit variations during different stages of resection. White circles = low potential outlier (more than 1.5 inter-quartile range at most 3 inter-quartile range.)

In a subgroup analysis, an attempt was made to highlight any differences in terms of iUS benefit by differentiating the ratings obtained for the three main categories of tumors: high grade gliomas, low grade gliomas and metastases (Figure 5). Among the various tumor categories, the Wilcoxon test with exact two-tailed significance did not produce significant differences for both 2D and 3D (Table A2).

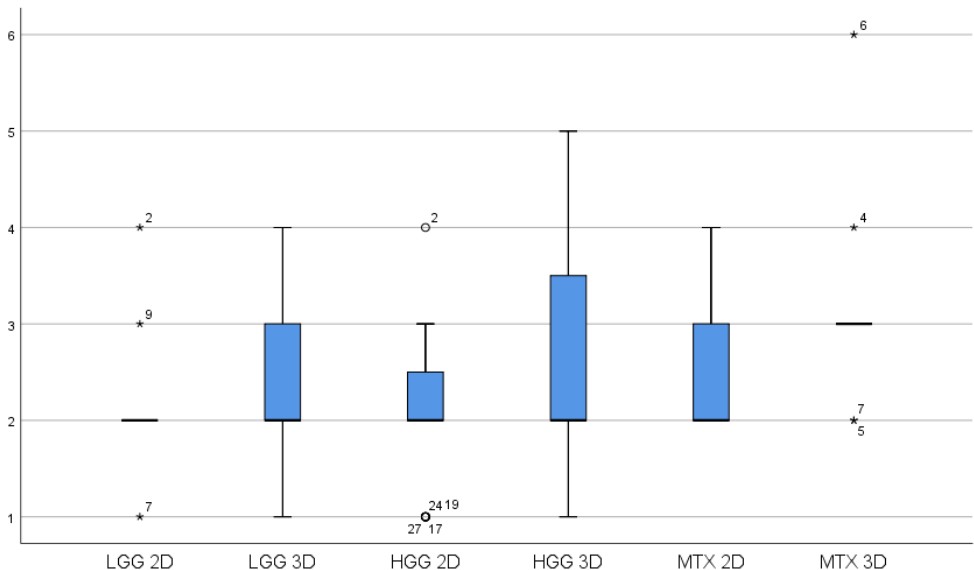

**Figure 5.** Box plot showing 2D and 3D benefit variations for the three main categories of tumors: HGG = high-grade gliomas, LGG = low-grade gliomas, MTX = metastases. White circles = White circles = low potential outlier (more than 1.5 inter-quartile range at most 3 inter-quartile range. * = high potential outliers of more than 3 inter-quartile ranges).

The accuracy of 2D and 3D navigation was judged on average as "good" in the three surgical phases, but negative ratings were recorded for 3D in a small number of cases due to the imprecision of the neuro-navigation software (Figure 6).

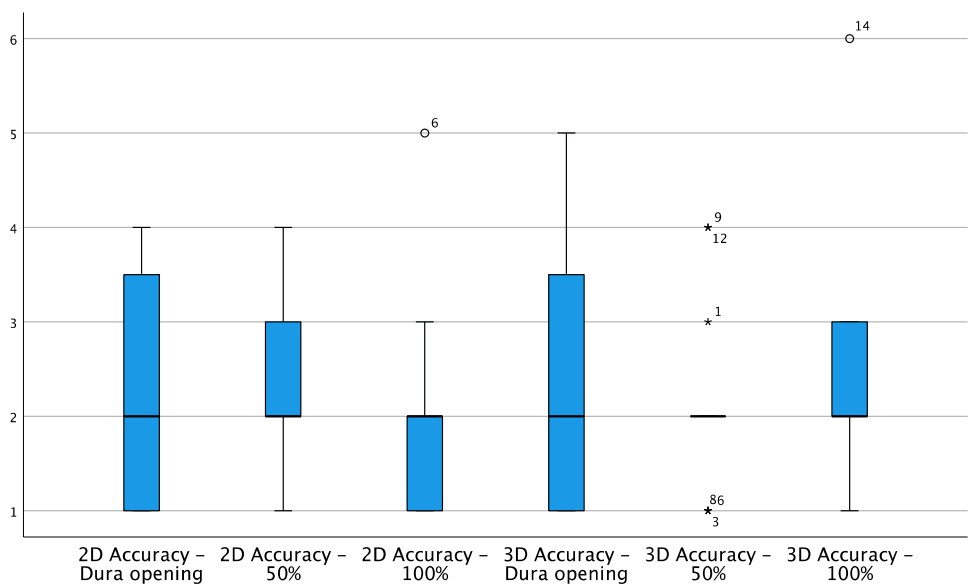

**Figure 6.** Box plot showing 2D and 3D accuracy variations during different stages of resection.

White circles = White circles = low potential outlier (more than 1.5 inter-quartile range at most 3 inter-quartile range.. * = high potential outliers of more than 3 inter-quartile ranges)

However, the statistical analysis performed did not show significant differences between the accuracy values for 2D and 3D in the various phases of the resection (Table A3), as it did not undergo significant variations from one phase of the resection to another.

The clinical values found in the postoperative period, collected from the third day onwards, were marginally worse than those recorded at the time of admission. The difference in the average scores of the NANO scale and KPS in the two periods, however, did not reach statistical significance (2.13 vs. 3.00, $p = 0.099$ for NANO scale; 83.13 vs. 75.63, $p = 0.061$ for KPS).

### 3.1. Illustrative Cases

#### 3.1.1. Case 1

The illustrative case of a 64-year-old patient presented with mild motor aphasia and right hemiparesis is reported below (preoperative NANO score = 4). Preoperative MRI with DTI sequences showed a left frontal expansive lesion of 78 cm$^3$, resulting in a mass effect on the ipsilateral ventricle (Figure 7).

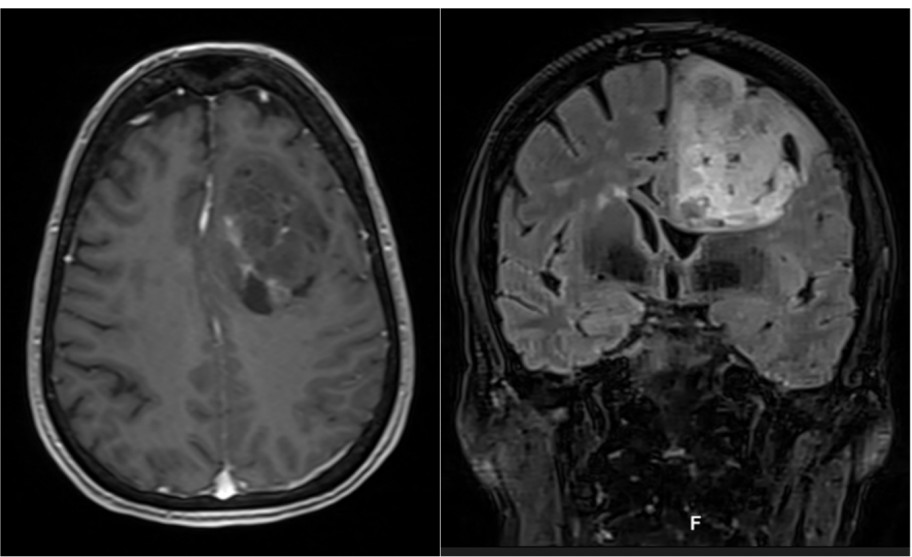

**Figure 7.** Axial contrast enhanced T1-weighted MRI (**left**) and coronal T2 flair (**right**) sequences showing a frontal left mass.

Surgical removal, performed with the aid of intraoperative neurophysiological monitoring, was completed and the patient experienced a temporary postoperative worsening of speech, which improved at discharge (postoperative NANO score = 6). The histological report documented a WHO II Oligodendroglioma. Table 1 shows the Likert scale-based ratings expressed by the surgeon in relation to the various stages of the surgery for the investigated parameters, namely, the image quality, benefit and accuracy of 2D and 3D iUS. Figures 8 and 9 show the iUS used in the two different stages of the surgery.

**Table 1.** Likert scale ratings reported for the frontal left WHO II Oligodendroglioma surgery presented in the text (1 = very good, 2 = good, 3 = fair).

| Stage of Surgery | 2D | | | 3D | | |
|---|---|---|---|---|---|---|
| | After Dura Opening | 50% Resection | 100% Resection | After Dura Opening | 50% Resection | 100% Resection |
| Image Quality | 1 | 2 | 3 | 2 | 3 | 3 |
| Benefit | 1 | 2 | 3 | 2 | 3 | 2 |
| Accuracy | 2 | 2 | 2 | 1 | 2 | 2 |

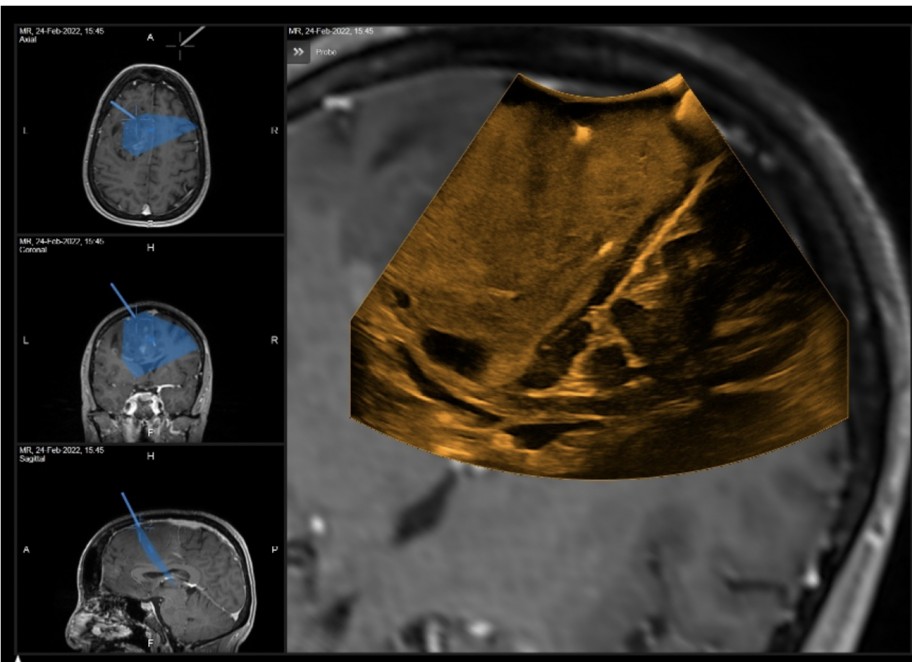

**Figure 8.** 2D iUS scan after dura opening.

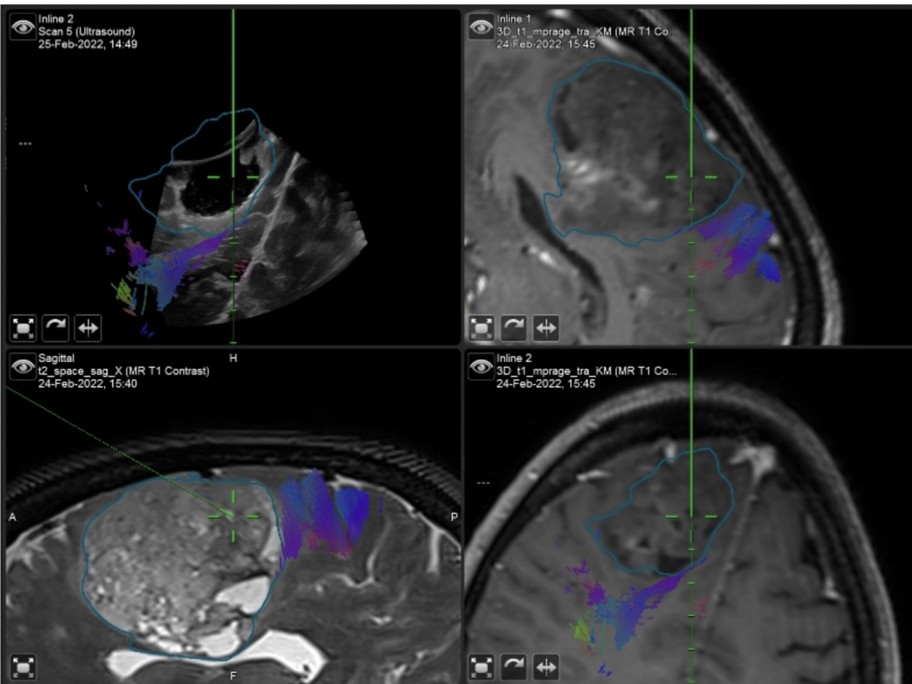

**Figure 9.** "live" iUS-guided resection before iMRI scan. Note how the cortical spinal bundle high-lighted in violet was spared from the resection area.

### 3.1.2. Case 2

A 67-year-old man underwent surgical resection of a right frontal glioblastoma (preoperative NANO score = 3). An initial neuronavigation showed inaccurate image matching between iUS and MRI (Figure 10). In this case, 2D iUS was more reliable in terms of tumor removal. However, this mismatch had no consequences on the postoperative surgical and clinical outcome (postoperative NANO score = 2).

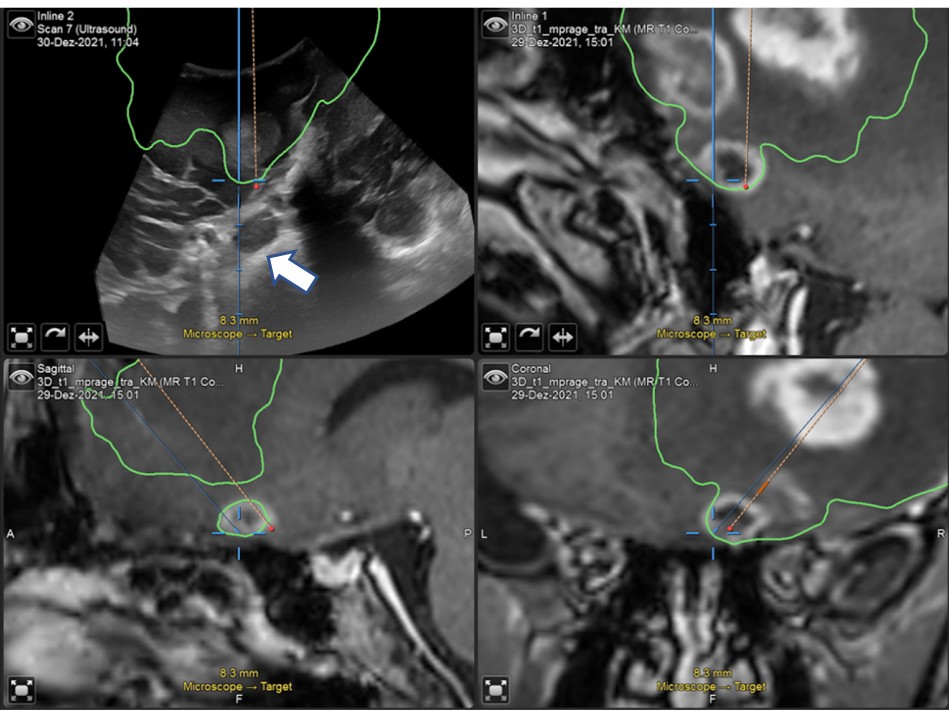

**Figure 10.** "live" iUS-guided resection at 50% of tumor removal showing the mismatch between the pointer and the real position on iUS (white arrow).

### 3.1.3. Case 3

A 68-year-old man presenting hemianopia and left hemiparesis associated with Gerstmann syndrome (preoperative NANO score = 4) underwent the surgical resection of a right parietal glioblastoma (Figure 11(left)). Although 2D and 3D iUS did not show suspected residual disease (Figure 12), iMRI documented a small residue (11 cm$^3$ out of an initial volume of 88.1 cm$^3$; Figure 11(right)), which was subsequently completely resected (postoperative NANO score = 2).

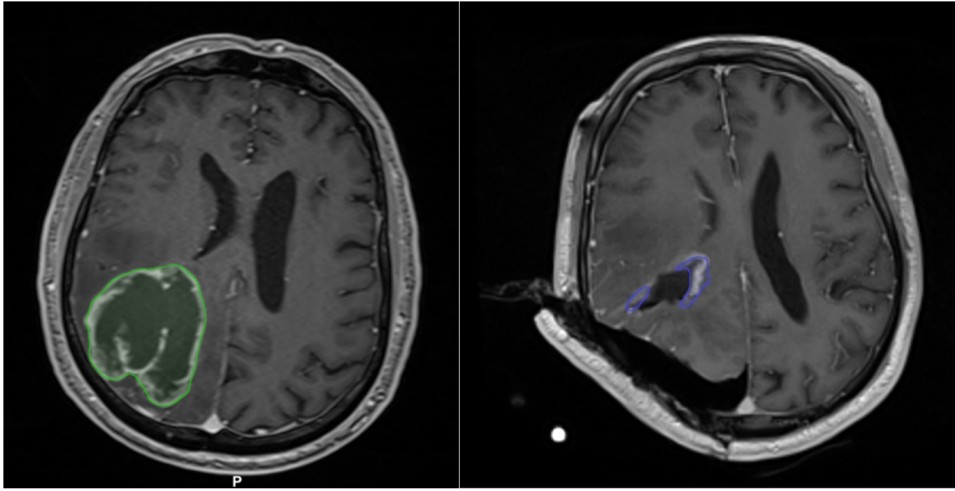

**Figure 11.** Preoperative (**left**) and intraoperative (**right**) MRI. Note the residual tumor outlined in violet along the edges of the surgical cavity on iMRI.

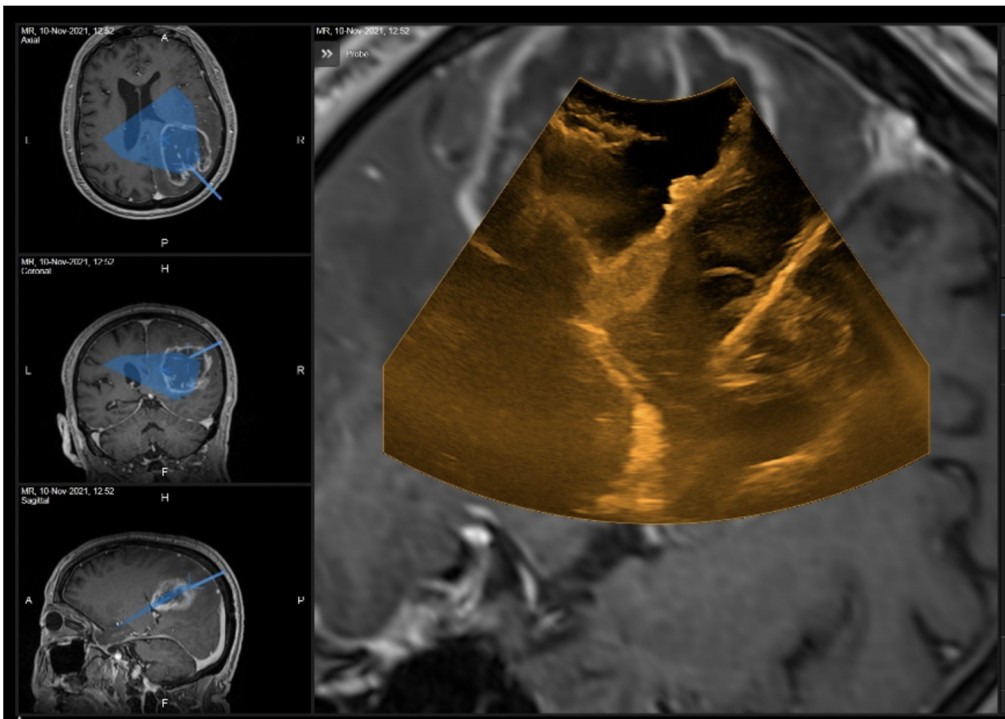

**Figure 12.** 2D iUS scan performed at the 100% surgical resection stage. No obvious residual disease was found.

## 4. Discussion

The prognostic importance of maximal safe tumor resection requires surgeons to take extra effort to achieve this goal. Intraoperative imaging is often an essential tool along this path. There is good evidence for use of iMRI to increase the extent of resection, both for high and low grade gliomas [26–33]. It allows for an intraoperative imaging update and provides images familiar to neurosurgeons from routine diagnostic scans. Nonetheless, various limits concerning the costs of installation and employment of dedicated technical staff are prohibitive in many hospitals. A further issue is a significant extension in operating times, which is on average around 60 min [33], as confirmed by our study (average iMRI time: 64 min). In terms of economy, iMRI is a highly inconvenient method for many centers.

However, a quick imaging update may often be necessary in particular situations, such as brain swelling caused by intracranial hypertension or awake surgery, to increase safety and accuracy. Iintraoperative ultrasound (iUS) allows for such an update as well as iMRI. iUS is available for neurosurgical use even longer than iMRI and 5-ALA. It has seen a resurgence of interest in recent years due to technological advancements, which overcame some relevant intrinsic issues. Among them, the suboptimal image quality of old ultrasound devices and presence of artifacts are now minimized by the possibility of using different types of probes at various frequencies [20,34]. In some systems, the probes can be dynamically modulated according to the depth of field [35]. The latest development of acoustic coupling fluids is also providing encouraging results [15,36,37].

Combining conventional iUS with neuro-navigation was an important step to overcome the orientation issue caused by oblique images with little overview provided by 2D iUS. The next step is calculating 3D volume models based on 2D iUS images. In this way, even the familiar three standard orthogonal planes can be provided in addition to an increased overview spanning the entire lesion [14].

The main features of interest in the use of navigated iUS, both in 2D and 3D, are the quality of the ultrasound images, the surgical benefit in terms of orientation and the control of the tumor and the accuracy of navigation. They were investigated in the present study in order to verify the differences between the two methods and to validate their use in

the surgical routine. The ratings of each of these investigated fields settled around the value of "2" of the Likert scale used in the study, corresponding to "good" (Figure 2). In particular, there was a tendency to "very good" ratings in terms of the quality of 2D iUS images, which were statistically superior compared to 3D iUS images. This difference most likely reflects the quality of the software interpolation of missing data between 2D slices during the scans. In our study, surgeons generally rated the quality of resulting 3D images. Quality was highest orthogonal to the scan direction and decreased when planes parallel to the scanning direction were depicted or in areas with a more curved scanning direction, resulting in a more fanned array of 2D slices in the scan. Hence, based on the tumor location and size, the positioning of patients and the scanning time during the course of surgery, a wide variety of imaging results can be expected. This being said, we found a relatively stable rating in terms of imaging results, even compared to scans before dural opening, involving a more linear method of scanning, and at the end of surgery, involving a curved scan following the form of the resection cavity.

It is also known that during the course of the resection the number of artifacts progressively increases at the bottom of the surgical cavity [15,23]. For this reason, one would have expected a deterioration in the quality, benefit and accuracy of the ultrasound images as the resection progresses both for 2D and 3D iUS. Surprisingly, the 2D and 3D quality remained unchanged on average during the resection, despite some worsening ratings in terms of the benefit and accuracy reported in the middle and at the end of the resection, with these differences not reaching statistical significance.

Solheim et al. investigated the quality of 2D iUS images through a questionnaire provided to the surgeon after high-grade glial tumor surgery [25]. A good or medium quality rating indicated an adequate image quality to correctly delineate the lesion borders. Consistent with our findings, such a result was reported in most cases (83%). Conversely, the iUS image quality was too poor to correctly define tumor borders in only 17% of cases. Reoperations and surgery in patients who underwent brain radiation therapy were significantly associated with poorer iUS image quality [25].

The perceived benefit of 3D ultrasound was rated as good before dural opening and in the middle of the resection, while it was rated 'fair' after complete tumor removal. Compared to an earlier series of our group comparing linear and sector array probes in a work flow assessment, high-resolution linear array probes reached a similar benefit rating of 2.3 as 3D reconstructed US in our series (2.0), while conventional sector array probes only reached a level of 3.8, showing a remarkable rating for an interpolated 3D reconstruction [34]. During surgery, the main benefit of 3D reconstructed iUS is the option to navigate in a virtual live ultrasound image. Hence, surgeons do not have to interrupt a resection for the purpose of observing the ultrasound depiction of a certain area. Further, it is easier to relocate areas of suspicion with regard to residual tumor, which can be very challenging, especially when using high-resolution probes with low penetration depth. With this in mind, it is not surprising that surgeons believe that they benefit most from 3D reconstructed iUS during resection for residual tumor control and as a compensation for brain shift. Meanwhile, after complete tumor resection, they relied more on the 2D scan. However, this might be only a subjective perception since only with a complete scan of the resection cavity could residual tumor be ruled out. Whether the whole resection cavity was covered can easily be determined by a 3D scan. On the other hand, especially at the end of resection, attenuation artifacts can be abundant. They can be more easily avoided or ruled out by scanning the same area from different angles, so more experienced ultrasound users might rely on a 2D scan in this situation.

Recently, Bastos et al. conducted a retrospective evaluation of patients undergoing image-guided tumor resection, in which both 3D iUS and iMRI were employed [38]. The authors retrospectively used a very similar routine compared to our prospective study protocol and the same ultrasound and navigation system. Similar to our study, scans were performed in three precise stages: before the dural opening, after the dural opening and immediately before the acquisition of the iMRI with which the iUS images were compared.

However, the authors focused on the comparison of iUS and iMRI for tumor detection. iUS correctly defined tumor localization and tumor borders and indicated relevant landmarks for guidance and orientation in nineteen out of twenty-three study participants (82.6%); the remaining four patients in which iUS was unable to detect a residual tumor were recurrence cases. As in our study, the issue of increased artifacts at the end of surgery was identified [38]. The use of ultrasound coupling fluid may overcome these types of artifacts and might even increase the value of 3D reconstructed iUS.

In recent years, research and technological development has led to growing evidence supporting the application of ioUS in order to obtain GTR in low- and high-grade gliomas. In particular, clinical studies are emerging to evaluate the impact of ioUS on EOR [38–42]. The next step of our ongoing prospective study will be to compare clinical and imaging data concerning EOR and outcome after the use of navigated 3D reconstructed ultrasound before intraoperative MRI compared to a matched pair retrospective group with neuro-navigation alone before iMRI. By this means, we hope to shed more light on how the extent of resection and the surgical workflow can be improved by navigated 3D iUS.

*Study Limitations*

The evaluations recorded concerning the benefit of 3D iUS in patients with brain metastasis presented a wide dispersion, reducing the reliability of the test and lowering the median score for this section compared to the others (Figure 5). Furthermore, as discussed in the case 2, some negative 3D accuracy ratings were recorded due to an imprecision in the primary registration of the patients, which resulted in the inaccurate spatial veracity of certain 3D images (Figures 6 and 10). This problem, in fact, was not related to the coupling of iUS with the Brainlab software, which was instead appropriate in all its phases, but was primarily detected in the 3D scan by the surgeon. For this reason, the questionnaire was slightly modified in its form during the study in order to more specifically judge in future how much the navigational accuracy is altered as a result of an anomaly in the ultrasound instrument rather than in the neuro-navigation software.

The experience of the surgeon in the use of iUS can affect both the usefulness of 2D and 3D images as well as 3D reconstructions results. Further limitations of this study include the small sample of patients from the prospective series and the subjectivity of the assessments provided by the surgeons. Since there was not the same rater for all surgeries, the expressed ratings could be influenced by the personal experience of each surgeon with iUS. No distinction was made between assessments based on the initial volume and eloquent or non-eloquent site of the tumor, and thus the limit deriving from the anatomical characteristics of resectability of the approached lesions was not considered. Furthermore, the limited number of patients did not allow for the performance of subgroup analysis for recurrent cases.

## 5. Conclusions

3D reconstructed iUS allows for an accurate intraoperative update of imaging with slightly lower image quality than 2D iUS. Yet, when used by experienced surgeons, it was still rated as "good" as was the perceived benefit of 3D iUS for the resection. The benefit for of tumor control and navigation accuracy were equally rated among 2D and 3D iUS navigation and did not undergo significant variations during tumor resection. The surgeons' subjective sense of the benefit in terms of tumor orientation and control was independent of the type of lesion and its grade.

In a future phase of this ongoing prospective study, the comparison of EOR and neurological outcome with a matched paired series of patients on whom the use of any imaging aids prior to iMRI was not performed could better help to evaluate the impact of navigated 3D iUS on a patient's outcome.

**Author Contributions:** Conceptualization, D.A. and J.C.; methodology, J.C. and D.A.; software, D.A. and J.C.; validation, D.A., A.P. (Andrej Pala) and J.C.; formal analysis, M.M.F., J.C., C.R.W. and A.P. (Andrej Pala); investigation, D.A., J.C., A.P. (Andreas Pfnür), Z.E. and A.P. (Andrej Pala); data curation, D.A., A.P. (Andreas Pfnür), Z.E. and J.C.; writing—original draft preparation, D.A., J.C. and A.P. (Andrej Pala); writing—review and editing, M.M.F., P.J.S. and C.R.W.; visualization, D.A., A.P. (Andrej Pala); supervision, J.C., M.M.F. and C.R.W.; project administration, J.C. and A.P. (Andrej Pala). All authors have read and agreed to the published version of the manuscript.

**Funding:** This research received no external funding.

**Institutional Review Board Statement:** The study was conducted according to the guidelines of the Declaration of Helsinki, and approved by the Ethics Committee of Ulm University (Nr. 283/19).

**Informed Consent Statement:** Written informed consent has been obtained from the patients in the study.

**Data Availability Statement:** More detailed data are available on demand after contacting the authors.

**Conflicts of Interest:** A.P. (Andrej Pala) and J.C. are working as Brainlab consultants.

## Appendix A

**Table A1.** Summary of patient descriptive data; * = missing data.

| # | AGE | Volume (cm$^3$) | Localization | Side | Eloquent Area | Recurrence | T OP (min) | T RM (min) | Diagnosis |
|---|-----|-----------------|--------------|------|---------------|------------|------------|------------|-----------|
| 1 | 62 | 35.6 | Temporal | Right | N | No | 255 | 45 | Metastasis |
| 2 | 53 | 2.11 | Temporal | Left | Y (speech) | Yes | 198 | 50 | Glioblastoma, IDH-wildtype |
| 3 | 33 | 1.57 | Frontal | Right | N | No | 247 | 60 | Astrocytoma IDH-mutant WHO II |
| 4 | 68 | 88.1 | Parietal | Right | Y (sight) | No | 386 | 70 | Glioblastoma, IDH-wildtype |
| 5 | 66 | 5 | Occipital | Left | Y (sight) | No | 226 | 70 | Metastasis |
| 6 | 82 | 40 | Temporal | Left | Y (speech) | No | 341 | 50 | Metastasis |
| 7 | 35 | 1 | Frontal | Right | N | Yes | 293 | 65 | Astrocytoma IDH-mutant WHO III |
| 8 | 62 | 33 | Temporal | Left | Y (speech) | No | 260 | 60 | Glioblastoma, IDH-wildtype |
| 9 | 67 | 35.3 | Frontal | Right | N | No | 243 | * | Glioblastoma, IDH-wildtype |
| 10 | 55 | 5 | Frontal | Right | Y (motor) | No | 358 | 60 | Astrocytoma IDH-mutant WHO IV |
| 11 | 77 | 35.5 | Frontal | Right | N | No | 300 | 60 | Glioblastoma, IDH-wildtype |
| 12 | 24 | 36.6 | Frontal | Right | N | No | 324 | 78 | Astrocytoma IDH-mutant WHO III |
| 13 | 66 | 56.1 | Temporal | Right | N | No | 392 | 60 | Glioblastoma, IDH-wildtype |
| 14 | 24 | 0.8 | Frontal | Right | N | No | 224 | 70 | Astrocytoma IDH-mutant WHO II |
| 15 | 56 | 59 | Frontal | Right | N | No | 405 | 87 | Oligodendroglioma WHO III |
| 16 | 64 | 78 | Frontal | Left | N | No | 474 | 77 | Oligodendroglioma WHO II |

**Table A2.** Descriptive statistics of 2D and 3D iUS benefit for the three main categories of tumors, with Wilcoxon Signed Ranks Test results; HGG = high-grade gliomas, LGG = low-grade gliomas, MTX = metastases.

| | | | | | | Percentiles | | | Wilcoxon Test | | |
|---|---|---|---|---|---|---|---|---|---|---|---|
| | N | Mean | Std. Deviation | Minimum | Maximum | 25th | 50th (Median) | 75th | HGG 2D-LGG 2D | MTX 2D-HGG 2D | MTX 2D-LGG 2D |
| LGG 2D | 9 | 2.22 | 0.833 | 1 | 4 | 2 | 2 | 2.5 | | | |
| HGG 2D | 27 | 2.07 | 0.781 | 1 | 4 | 2 | 2 | 3 | *p* = 1.000 | *p* = 0.375 | *p* = 0.500 |
| MTX 2D | 9 | 2.56 | 0.726 | 2 | 4 | 2 | 2 | 3 | | | |
| | | | | | | | | | HGG 3D-LGG 3D | MTX 3D-HGG 3D | MTX 3D-LGG 3D |
| LGG 3D | 9 | 2.33 | 0.866 | 1 | 4 | 2 | 2 | 3 | | | |
| HGG 3D | 27 | 2.48 | 1.252 | 1 | 5 | 2 | 2 | 4 | *p* = 0.313 | *p* = 0.938 | *p* = 0.078 |
| MTX 3D | 9 | 3.22 | 1.202 | 2 | 6 | 2.5 | 3 | 3.5 | | | |

**Table A3.** Accuracy of iUS raw data with Wilcoxon Signed Ranks Test. * = missing data.

| # | 2D Accuracy—Dura Opening | 3D Accuracy—Dura Opening | 2D Accuracy—50% | 3D Accuracy—50% | 2D Accuracy—100% | 3D Accuracy—100% |
|---|---|---|---|---|---|---|
| 1 | 4 | 4 | 3 | 3 | 3 | 3 |
| 2 | 1 | 1 | 2 | 2 | * | * |
| 3 | 1 | 1 | 2 | 1 | 1 | 3 |
| 4 | 2 | 2 | 2 | 2 | 1 | 1 |
| 5 | 3 | 3 | 3 | 2 | 2 | 2 |
| 6 | 1 | 1 | 1 | 1 | 5 | 2 |
| 7 | 1 | 1 | * | * | 3 | 3 |
| 8 | 3 | 2 | 1 | 1 | 2 | 2 |
| 9 | 4 | 4 | 2 | 4 | 1 | 1 |
| 10 | 2 | 5 | * | * | 2 | 2 |
| 11 | 1 | 1 | 2 | 2 | 1 | 1 |
| 12 | 4 | 3 | 4 | 4 | * | 3 |
| 13 | 4 | 2 | 3 | 2 | 2 | 2 |
| 14 | 2 | 4 | * | * | 2 | 6 |
| 15 | 2 | 1 | 2 | 2 | 2 | 2 |
| 16 | 2 | 1 | 2 | 2 | 2 | 2 |
| Wilcoxon Test | *p* = 0.813 | | *p* = 1.000 | | *p* = 0.750 | |

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
