# Peer review of "Evaluation of a Navigated 3D Ultrasound Integration for Brain Tumor Surgery: First Results of an Ongoing Prospective Study"

_curroncol, doi:10.3390/curroncol29090518_

Round 1
Reviewer 1 Report
The authors present a study on the applicability of 3D ultrasound for the evaluation of brain tumors. The topic is interesting and topical, especially since in this field important steps are being taken for the use of ultrasound for the treatment of some brain tumors but also for their evaluation.
The subject addressed is novel from the point of view of clinical tests because no medical results have been reported in the specialized literature, the lack of technology being presented from a mathematical point of view. Since this technology is at the beginning of the development path, the authors can collaborate with different academic or private research institutes for the development of a hardware and software technology that will improve the performance of medical analysis for the detection and treatment of brain tumours.In order to reduce the limitations specified in the study, I recommend the authors to open a research project for the development of software applications that can increase the accuracy of cerebral tumorilot detection using ultrasound. Currently, in such applications, complex algorithms can be used that allow the restoration of images from ultrasound-type signals, and from a hardware point of view, an area of ultrasound sensors can be created that realizes beamforming focused on a certain brain area.
If the authors are interested, we can discuss this topic.
Author Response
Dear reviewer,
thank you very much for the valuable review. The proposal for the development of a software application to increase the diagnostic accuracy of the intraoperative ultrasound technique certainly appears interesting to our group. We would surely be interested in discussing this topic in the future.
In this regard, a collaboration with Brainlab is underway for the development of a non-rigid registration algorithm of intraoperative 3D ultrasound, also using data collected in this study. This can be very useful in providing automatic update of preoperative neuronavigation imaging through the use of intraoperative ultrasound.
Reviewer 2 Report
The manuscript provides first results on a very interesting topic that is of high relevance in neurosurgical practice.
The introduction provides sufficient background on the topic. However, there are several recent publications on the use of (navigated) intraoperative ultrasound and extent of resection that might be worth mentioning here or in the discussion section to provide a broader overview of recent work.
Within the methods section the authors should provide more details on the application of iUS at different stages during surgery. Which probe was used? Were the same settings used for all cases or were any specific adaptions made? Did 2D scans follow a specific orientation (approximately axial/sagittal/coronar depending on the tumor location or in an oblique direction)? How was a 3D iUS data set acquired (single sweep, multiple sweeps, orientation)? In case of navigated iUS (see Figure 8), was only the overlay used or also presegmented tumor outlines displayed in the 2D mode to evaluate tumor location / shift etc.?
The results should include more illustrative cases. In case of the presented illustrative case Figure 8 and 9 could be improved by displaying 2D and 3D iUS at the same stage of surgery (to compare both modalities) as well as over the course (corresponding 2D and 3D) to provide the reader with more information about the image quality, that is underpinned by the surgeons ratings. An additional case should be added providing a case where there were some problems or issues such as detection of navigational inaccuracies (as mentioned for one case in the Limitations section), to provide the user also with an example of its capability to identify those inaccuracies. Maybe also a case with a remnant, seen on iUS and iMRI, or not seen on iUS but on iMRI might underpin the potential of iUS usage.
How many surgeons did perform iUS in those cases? Was there any difference seen between the different surgeons?
Author Response
Dear reviewer,
thank you for the comments and recommendation which helped to improve the value of our manuscript.
As recommended, references to some recent articles which appeared to be consistent with the discussion have been added. The topic of the extent of resection, as reported in the text, will be dealt with more extensively in the continuation of the ongoing clinical study.
More details on the application of iUS have been provided in methods, as suggested.
5-13 MHz micro-convex transducer was used in all surgical steps. The same setting was used for all cases. 2D scan did not follow a specific orientation but varied according to tumor location. 3D dataset was acquired by a single wide sweep, following the orientation that ensured greater tumor coverage through craniotomy. Furthermore, we used presegmented tumor outlines mainly to evaluate orientation.
It was already our intention to display 2D and 3D iUS images at the same stage of surgery, in order to provide visual feedback of the difference in image quality. Unfortunately, the images collected during the study could not fulfill this purpose.
However, 2 more illustrative cases were added. One case where an inaccuracy problem emerged as a result of an equipment registration defect. The other case concerns the finding of a residual tumor that could not be diagnosed with iUS, but only with iMRI. In this case, in fact, the artifacts along the edges of the removed tumor hindered the finding of the residue.
Four surgeons with experience in the use of neuronavigation and iUS were involved in the study. Differences between surgeons in the use of iUS could not be assessed, mainly due to the small number of patients enrolled. We will include this question in further studies as an important and interesting additional subject.
Round 2
Reviewer 2 Report
All aspects were adressed sufficiently.